# Selected Patients with Unresectable Perihilar Cholangiocarcinoma (pCCA) Derive Long-Term Benefit from Liver Transplantation

**DOI:** 10.3390/cancers12113157

**Published:** 2020-10-27

**Authors:** Adiba I. Azad, Charles B. Rosen, Timucin Taner, Julie K. Heimbach, Gregory J. Gores

**Affiliations:** 1Division of Gastroenterology and Hepatology, Mayo Clinic, Rochester, MN 55905, USA; azad.adiba@mayo.edu; 2Division of Transplantation Surgery, Mayo Clinic, Rochester, MN 55905, USA; rosen.charles@mayo.edu (C.B.R.); taner.tumucin@mayo.edu (T.T.); heimbach.julie@mayo.edu (J.K.H.)

**Keywords:** chemoradiation, primary sclerosing cholangitis, organ allocation

## Abstract

**Simple Summary:**

Cancers affecting the bile duct may occur in different anatomic regions of the bile duct and are called cholangiocarcinomas. A subset of these cancers involve the bile ducts just below the liver in a region named the hepatic hilum and are referred to as perihilar cholangiocarcinomas. These small cancers are often unresectable (cannot be taken apart by surgery), as they include crucial blood vessels and other bile ducts of the liver. Herein, we report on the role of liver transplant in these patients, which can lead to long term disease free survival.

**Abstract:**

Selected patients with unresectable perihilar cholangiocarcinoma (pCCA) derive long-term benefits from liver transplantation. Between 1993–2019, our group at Mayo Clinic performed 237 transplants for pCCA. With this experience, we note that two distinct patient populations comprise this group of pCCA patients: those with underlying primary sclerosing cholangitis (PSC) and those without identifiable risk factors termed sporadic or de novo pCCA. Long-term survival after transplant is better in PSC patients (74% five-year survival) than in those with de novo pCCA (58% five-year survival). Herein, we review the likely clinical factors contributing to the divergence in outcomes for these two patient populations. We also offer our insights on how further advances may improve patient selection and survival, focusing on the de novo pCCA patient population.

## 1. Introduction 

Cholangiocarcinoma (CCA) is a hepatobiliary malignancy with features of biliary epithelial cell differentiation. It is the second-most common primary hepatobiliary malignancy after hepatocellular carcinoma (HCC). Worldwide, the incidence of all cholangiocarcinomas has increased over the last 40 years [1]. Anatomically, CCA may be classified as intrahepatic, iCCA (proximal to the second degree bile ducts), perihilar, pCCA (above the cystic and below the second order bile ducts), and distal, dCCA (between the cystic duct and the ampulla of Vater). Based on the macroscopic appearance of the tumor, cholangiocarcinomas may be classified as mass-forming, intraductal-growing, and periductal-infiltrating [2]. Histopathologically, CCA may contain mucin-producing or non-mucin-producing glands, and most tumors consist of a mix of these features. CCA pose a therapeutic challenge due to their genetic heterogeneity, dense stroma surrounding tumor cells impeding diagnoses by biliary brushings, and a densely populated tumor microenvironment. The anatomic subtypes have their unique molecular biology, natural history, and prognosis and, therefore, require distinct treatment approaches. Surgery is the most effective treatment modality for all anatomic subtypes of CCA. Unfortunately, less than a third of patients present early enough to be eligible for curative surgery [3]. A select subset of patients with unresectable pCCA qualifies for liver transplantation. Analogous to locally advanced pancreatic cancers, these patients have unicentric cancers that are not anatomically resectable but do not have evidence for intrahepatic or extrahepatic metastasis. Liver transplantation for small iCCA has been performed but is controversial and is not a standardized indication. In this review, we focus on the treatment of de novo pCCA with liver transplantation (as described in the subsequent section) and discuss potential avenues for improvement in the management of this complex and challenging disease.

## 2. Diagnosis of pCCA and Candidacy for Liver Transplantation

Perihilar CCA was first described by Gerald Klathskin in 1965 [4] and is the most common (50%) of all CCA but in itself is a rare malignancy, with 5000–7000 cases diagnosed every year in North America [5]. Most cases of pCCA are sporadic or de novo. Established risk factors for pCCA are primary sclerosing cholangitis, PSC (which carries a 5–10% lifetime risk of developing CCA), liver fluke infection, and hepatolithiasis. 

PSC is a chronic biliary tract disease characterized by inflammatory, diffuse biliary strictures of the extra- and intrahepatic biliary tree. Over time, the resulting cholestasis leads to parenchymal liver injury, cirrhosis, and portal hypertension. End-stage liver disease due to PSC is a frequent indication for liver transplantation. PSC may be viewed as a preneoplastic liver disease, as the cumulative incidence of CCA is approximately 25% over twenty years for men and 15% for women [6]. The vast majority of these CCA are pCCA. Given this risk for CCA, current guidelines recommend annual surveillance studies with MRI/MRCP (magnetic resonance cholangiopancreatography) and serum CA 19-9 determinations in PSC patients [7]. Hence, pCCA in PSC is often diagnosed during surveillance and is more likely to be diagnosed at an earlier stage than in patients without PSC. pCCA associated with PSC is deemed unresectable due to the presence of parenchymal liver disease, skip lesions in this cancer, which are common, and the biliary field defect, which makes metachronous lesions common over time. In contrast, patients with de novo pCCA often present with painless jaundice, a liver mass, and are often unresectable due to bilateral bile duct extension and the vascular encasement of contralateral vessels if the disease predominantly arises in either the right or left hepatic ducts. 

The diagnosis of pCCA utilizes a series of laboratory, radiographic, and endoscopic tests. In brief, the diagnosis is made on a malignant-appearing hilar stricture or mass, elevation in the primary serum biomarker for CCA, cancer antigen 19-9 (CA 19-9) >130 U/mL, and cytology and chromosomal aberrations (using fluorescent in situ hybridization, FISH) present in samples obtained from biliary brushings during endoscopic retrograde cholangiopancreatography (ERCP). Notably, 7% of the population may have a falsely normal or low CA19-9 due to being Lewis antigen nonproducers and, therefore, lack the ability to secrete CA 19-9. High-quality, cross-sectional imaging with arterial and porto-venous phases are key in determining the location and extent of hilar tumors, vascular involvement, and the presence of intrahepatic or localized metastasis. Although Liver Imaging Reporting and Data System (LI-RADS) criteria have been reported for intrahepatic CCA (iCCA), they have not been developed for pCCA. Indeed, pCCA usually presents as an obstructive biliary stricture (painless jaundice), with proximal bile duct dilatation without a mass lesion. As LI-RADS criteria are developed for well-defined mass lesions, it is unlikely that LI-RADS criteria for pCCA will ever be developed. Of note, since these tumors are often not fludeoxyglucose (FDG)-avid, the sensitivity of positron emission tomography (PET) is low. Endoscopic ultrasound (EUS)-guided fine needle aspirates of peri-hepatic lymph nodes are used to determine whether the tumors are localized to the biliary system or display an extrahepatic spread. Since patients with pCCA present with obstructive jaundice, ERCP acts as both a diagnostic and therapeutic intervention. However, due to the desmoplastic (fibrotic) nature of these tumors, endobiliary brushings or biopsies provide a definitive diagnosis in only 50% of cases. Cytologic assessment from endobiliary brushings are comprised of conventional cytology and FISH. Conventional cytology may find cells that are negative, atypical, suspicious, or positive for adenocarcinoma. Sampling errors, a lack of adequate cell numbers, and the subtle differences between malignant and benign cells are some of the challenges associated with conventional cytology. Furthermore, PSC or acute cholangitis may result in reactive cells that may appear identical to malignant cells. As a result, conventional cytology on its own has a low sensitivity rate in detecting pCCA (15%) when the samples used are only positive for malignancy and 38% when samples that have both suspicious and malignant cells are used [8]. The introduction of FISH has enhanced the sensitivity of cytology. FISH detects structural chromosomal abnormalities (a hallmark of cancer) in cells using fluorescently labeled DNA probes. The DNA probes used in the diagnosis of pCCA by FISH are specific for the pericentromeric regions of chromosomes 3, 7, and 17 and the chromosomal locus 9p21. The results of the FISH analyses can be categorized as negative or disomic (two copies for each probe), trisomic 7 (≥10 cells with three copies of chromosome 7 and two or fewer copies of the other three probes), or polysomic (≥five cells with ≥ three signals in two or more of the four probes). FISH polysomy, in combination with atypical or suspicious appearing cells, are strongly suggestive of pCCA [9,10]. Furthermore, percutaneous and laparoscopic biopsies are not recommended, as they predispose the patient to developing metastasis, rendering them ineligible for liver transplantation [11]. Patients with de novo pCCA first undergo a rigorous assessment to determine whether they are eligible for resection, which is the standard treatment. It is currently not known whether surgical resection in this group of patients (de novo pCCA, which are resectable, small, and without locoregional spread) offers the same outcome as liver transplantation after chemoradiation, as some groups observe superior outcomes with transplantation [12] and others observe no difference in outcomes between the two approaches [13]. Neoadjuvant chemoradiation therapy followed by liver transplantation is the standard of care for the management of localized pCCA, which is considered to be unresectable. 

The goal for neoadjuvant chemoradiation and liver transplantation for unresectable pCCA is to achieve R0 margins and complete pathologic response (i.e., no residual tumor on explant). In 1994, Alden et al. reported a two-year survival benefit of 48% vs. 0% in 48 patients with extrahepatic CCA receiving high- vs. low-dose radiation and brachytherapy [14]. Survival was observed to be even greater when high-dose chemoradiation and brachytherapy was combined with a simultaneous infusion of 5-fluorouracil (5-FU) [15]. The Nebraska liver transplant group first combined neoadjuvant chemoradiation and liver transplantation for a selected group of patients with pCCA (half of whom had de novo CCA) and observed encouraging long-term survival rates, but this approach was fraught with brachytherapy-related sepsis and bile duct perforation and postoperative complications [16]. Subsequently, our group at the Mayo Clinic formalized a protocol for a rigorously selected group of patients with unresectable pCCA and reported very favorable long-term outcomes following neoadjuvant chemoradiation and liver transplantation [17,18]. The critical factors in determining the transplant candidacy are the tumor size (radial tumor diameter of 3 cm or less) and unresectability assessed by bilateral involvement of the bile ducts such that biliary reconstruction is not feasible, unilateral lobar atrophy with contralateral bile duct involvement, and unilateral bile duct involvement with contralateral encasement of the hepatic artery. Patients are excluded from a transplant if they display intra- and extra-hepatic metastasis, prior abdominal surgeries that may have violated the tumor plane, and other comorbidities that impair their functional status. Once the patient is selected for liver transplantation with neoadjuvant therapy, they are listed on the United Network for Organ Sharing (UNOS) and assigned MELD exception points. Neoadjuvant therapy comprises external beam radiotherapy (EBRT) given in 30 fractions twice daily for three weeks for a total dose of 4500 cGy with a continuous infusion of 5-FU administered for the duration of the EBRT. High-dose brachytherapy administered intraductally by ERCP is delivered through transcatheter iridium-192 seeds. Patients then take oral capecitabine (2000 mg/m^2^ of body surface area) until the time of transplantation. Near the time of transplantation, patients undergo a hand-assisted laparoscopic staging operation with a biopsy of any suspicious lesions and periportal lymph nodes to exclude metastasis. We emphasize that the decision to resect or transplant is critical and must be made before any therapy is pursued, because the initiation of one approach excludes the other. Patients who have undergone a resection attempt with dissection of the hilum of the liver cannot be considered for liver transplantation, as the tumor plane has been violated, leading to a high chance of seeding. Similarly, patients who have undergone chemoradiation treatment in anticipation of liver transplantation cannot undergo resection due to the degree of inflammation and damage sustained by the perihilar area. In summary, the diagnosis, staging, and management of pCCA requires a thoughtful multidisciplinary approach consisting of expert hepatologists, advanced endoscopists, pathologists, and hepatobiliary/transplant surgeons. 

## 3. Outcomes of Liver Transplantation for De Novo Perihilar CCA: Mayo Clinic Experience 

Between 1993–2019, 376 patients were enrolled in the CCA transplant protocol, with a 14% drop-out rate due to disease progression. One hundred and forty-eight of these patients were enrolled with de novo pCCA and experienced a similar rate of drop-out (14%) due to disease progression prior to the staging surgery. The de novo group had one patient with alcoholic liver disease, one patient with hepatitis B virus (HBV), two patients with hepatitis C virus (HCV) and one patient with HCV and human immunodeficiency virus (HIV) coinfection, and four patients with non-alcoholic steatohepatitis (NASH). Of the patients with de novo pCCA who underwent the staging surgery (*n* = 117), 26% had advanced disease diagnosed during the operation, with peritoneal and regional lymph node metastases being the two most common sites. This is in contrast to patients who underwent a staging operation for PSC-related pCCA, where 14% of the patients staged positive. Therefore, patients with de novo pCCA experience a higher rate of drop-out at staging prior to liver transplantation than patients with PSC-associated pCCA. As of 2019, 237 patients (with de novo and PSC-related pCCA) underwent liver transplantation, where two-thirds of the patients received a deceased donor liver. Overall, the intention-to-treat analysis revealed that the five-year survival for all patients after transplantation was 68% ± 3% and, at 10 years, 60% ± 4% (Figure 1). In total, 84 patients with de novo pCCA underwent liver transplantation from 1993–2019, and the majority (70%) received deceased donor livers. Patient survival in the de novo group at five years was 58% ± 6% and 47% ± 6% at 10 years after transplant (Figure 2). This is significantly lower than the survival observed in patients with PSC-associated pCCA, where the five and 10-year survivals were 74% ± 4% and 67% ± 4%, respectively (*p* = 0.02) (Figure 2). The lower survival rate is primarily driven by the higher risk of recurrence after liver transplantation in the de novo vs. the PSC patients, 45% ± 6% vs. 22% ± 4% at five years, respectively (*p* = 0.001) (Figure 3). However, we have also demonstrated that the superior survival for PSC patients is likely due to having a less advanced disease at the time of enrollment. When controlled for risk factors such as size of the tumor, CA 19-9 level, portal vein encasement, and amount of residual tumor in the explant, which may be impacted by both the tumor burden and treatment responsiveness, the risk of disease recurrence for PSC and non-PSC patients is noted to be similar [19,20]. Between 1993–2019, the disease recurrence risk for all patients was 30% ± 3% at five years and was observed primarily in the abdomen, followed by chest. A diagnosis of recurrence was made at an average of 26 months after liver transplantation (range three months to 11 years), similar to what has been reported before [21]. 

Factors associated with recurrences of CCA have evolved over the years as transplant centers have gained more experience and, therefore, modified the selection and treatment protocols. Tumor recurrence after transplant is almost always outside of the biliary system. Transperitoneal tumor biopsy and mass size greater than 3 cm in radial diameter are important predictors in tumor recurrence [21,22]. Portal vein encasement, which reflects the extent of tumor burden, and the absence of oral capecitabine therapy prior to transplant were associated with residual tumor on the explant, which, in turn, was associated with recurrence [19,20]. Multiple studies have shown the presence of a residual tumor to be associated with a higher rate of recurrence and reduced survival [22,23]. In particular, Lehrke et al. closely examined the extent of residual tumors and other histopathologic factors in predicting outcomes in a detailed study [24]. A residual tumor was estimated by a histologic assessment of the tumor bed area, where the relative magnitude of the residual viable tumor to treatment-related necrosis and fibrosis was measured. The overall percentage of the residual viable carcinoma was calculated as the average of the estimated viable carcinoma percentages in all slides and were named as the estimated residual tumor (ERT) and then classified into four categories: (1) (complete/near-complete response: ≤1% ERT), (2) (marked response: >1 to <10% ERT), (3) (moderate response: 10 to <30% ERT), and (4) (minimal response: ≥30% ERT). Category 1 included cases with no residual carcinoma or with only rare isolated tumor cells scattered in the tumor bed. A five-year survival rate for patients for ERT 1 was 57% vs. 9% in ERT 4. It is important to note, however, that the absence of a residual tumor on the explant does not eliminate the possibility of recurrence but does lower the risk [19,20,22,24]. The histopathologic characteristics of the explant are also important in predicting tumor recurrence [22,24]. In particular, the presence of perineural invasion was associated with a significant risk of recurrence after adjusting for confounding factors in a multivariate analysis [22,24]. Lehrke et al. also examined the predictive power of the histological grade of the residual tumor in the explant. The four-tier grading system, which is based on the percentage of the glandular component of tumors, was used to determine the histologic grades of the tumor (progressive glandular loss denotes a loss of differentiation): grade 1—well-differentiated, grade 2—moderately differentiated, grade 3—poorly differentiated, and grade 4—undifferentiated. Interestingly, the histologic grade of the residual tumor on the explant was not a significant predictor of recurrence [24]. 

## 4. Way Forward 

Although the treatment protocol provides survival benefits in a disease that otherwise has a 50–70% mortality if untreated in two years, we are still faced with challenges of drop-out during the treatment and tumor recurrence after liver transplantation. These challenges are crucial to resolve, as the organ pool is significantly small compared to the demand, and the chemoradiation treatment is aggressive and potentially toxic. Factors such as advanced age, prior cholecystectomy, and mass identified on imaging were initially concluded to significantly influence treatment morbidity and post-transplant recurrence, based on the early Mayo Clinic experience [22]. Subsequently as we managed a larger cohort of patients, we were able to conduct a more robust data analysis and observed that factors such as a radial diameter of tumor ≥3 cm and CA 19-9 of 500 ≥ IU/mL in the absence of cholangitis portended a worse prognosis [19]. As our experience at the Mayo Clinic in treating this unique malignancy has matured with the increasing patient numbers, we question whether de novo pCCA should be treated as a distinct malignancy rather than PSC-associated pCCA. Currently, patient selection/exclusion criteria are almost identical in both patient groups. However, with regards to patients with de novo pCCA, there is a high drop-out rate and less than desired post-transplant survival rate. Therefore, we discuss the following issues that could be modified to refine the protocol to lower the drop-out rates after entering the protocol and to achieve higher rates of disease-free survival after live transplantation in patients with de novo tumors. Approximately 70% of patients with de novo pCCA present with a mass on diagnosis, compared to 47% in PSC patients [19]. This is due to the frequent and close monitoring of PSC patients with ERCP-guided biliary brushings often leading to the diagnosis of cholangiocarcinoma based on chromosomal changes indicative of malignancy or high-grade dysplasia (i.e., FISH polysomy). Furthermore, the protocol currently allows for a radial diameter of 3 cm, with no restrictions on the longitudinal diameter. Perhaps restricting these parameters to smaller dimensions in patients without PSC would result in a selection of earlier stage tumors. In fact, one may argue that a cholangiocarcinoma that presents with a mass contains malignant cholangiocytes that have already acquired the ability to invade the hepatic parenchyma, as opposed to tumors that are periductal-infiltrating (extend along the wall of the bile duct) and intraductal-growing (grow intraluminally). The relationship between unilateral lobar atrophy and contralateral vascular involvement should also be considered. The two primary etiologies of lobar atrophy are the obstruction of the portal blood and bile flow. In the absence of PSC, lobar atrophy is likely from a combination of malignant obstruction of the portal vein and bile ducts, suggesting the presence of the tumor for a long time and, therefore, more opportunity for malignant cells to infiltrate the lymphatic system. Furthermore, unilateral lobar atrophy with contralateral vascular involvement in such patients likely represents an advanced and/or aggressive tumor that may not be curbed adequately by neoadjuvant chemoradiation therapy. Occlusion of the hepatic blood vessels is generally not considered to be a contraindication to the CCA treatment protocol, as long as it does not a pose a surgical barrier during transplant. The encasement of a blood vessel may represent a disease that is likely to seed (or that has already seeded) in the lymphatic system and be potentiated by immune-suppression following transplant, thereby being predisposed to a high risk of tumor recurrence. Patients with biliary obstruction who have complex or challenging anatomy or who are managed at centers without experienced advanced endoscopists may require percutaneous transhepatic biliary catheters for biliary decompression. These procedures may result in the seeding of malignant cells in the peritoneum. Another critical factor in determining patient outcomes may be in how the tumor responds to chemoradiation therapy. Unfortunately, a radiologic evaluation is limited due to inflammation and necrosis of the hilum in response to chemoradiation, rendering it impossible to determine the presence of a viable tumor. It may be that pCCA occurring in the setting of PSC is more responsive to radiation therapy, though there is no direct evidence to support this concept. Furthermore, due to limitations in accessing and sampling pCCA, there is no pathologic data available until the time of transplant with regards to tumor reduction in response to chemoradiation. Noninvasive techniques such as cell-free DNA methylation assays or next-generation sequencing (NGS) of circulating cell-free DNA analysis have the potential to detect micrometastases, which complicate the disease course of pCCA. Finally, considerations may be made in individualizing the chemoradiation portion of the protocol to patient characteristics. All patients on the protocol receive a standard dose of radiation of 4500 cGy over 30 fractions. Since patients with de novo perihilar CCA do not have the problem of chronic biliary strictures and inflammation, such as patients with PSC, they may tolerate a higher and perhaps more effective dose of radiation therapy without experiencing increased adverse events related to radiation-induced damage. 

In the current organ allocation system, patients with hepatocellular and pCCA are treated in an identical fashion in terms of receiving MELD exception points. However, a recent study focused on the waitlist drop-out and post-transplant graft survival rates for both hepatocellular carcinoma (HCC) and pCCA based on the Scientific Registry of Transplant Recipients (SRTR). When matched for the Organ Procurement and Transplantation Network (OPTN) region and listing date, patients with pCCA had a significantly higher waitlist drop-out rate (six- and 12-month drop-out rates of 13.2% and 23.9%, respectively) than patients with HCC (six- and 12-month drop-out rates of 7.3% and 12.7%, respectively) [25]. Furthermore, transplants for pCCA were associated with lower rates of graft survival than with HCC: 60.7% vs. 81.6% at three years, respectively [25]. The recurrence of CCA is most likely what is driving the higher risk of graft loss in patients who undergo transplant for pCCA, as perioperative vascular complications (which are likely associated with radiation injury) are commonly managed without the consequence of graft loss [23]. These striking statistics underscore the importance of developing strategies to accurately identify and predict the biologic behavior, response, or sensitivity to neoadjuvant chemoradiation and the true extent of pCCA. 

## 5. Conclusions

Neoadjuvant chemoradiation and liver transplantation is the only curative treatment for patients with pCCA who meet specific criteria for the treatment protocol and leads to much higher long-term survival than either treatment alone. However, despite significant advances in the field, patients with de novo pCCA have a lower than desired post-transplant survival primarily due to high rates of disease recurrence, as showcased in this review. Future strategies will need to incorporate more effective neoadjuvant therapy. 

## Figures and Tables

**Figure 1 cancers-12-03157-f001:**
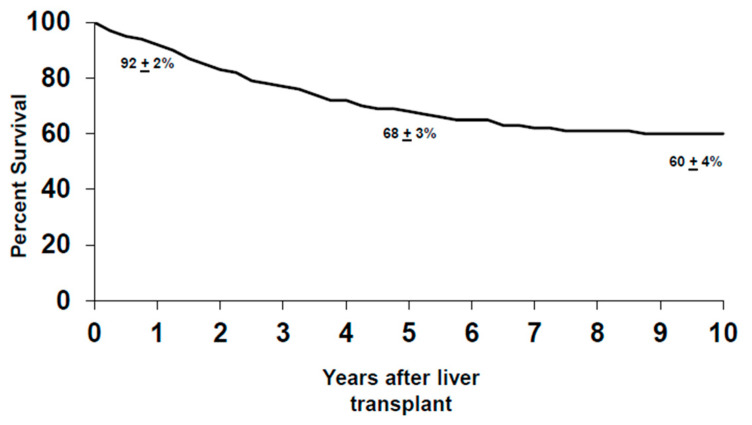
Intention-to-treat analysis of the overall survival of patients with perihilar cholangiocarcinoma (pCCA) after liver transplantation.

**Figure 2 cancers-12-03157-f002:**
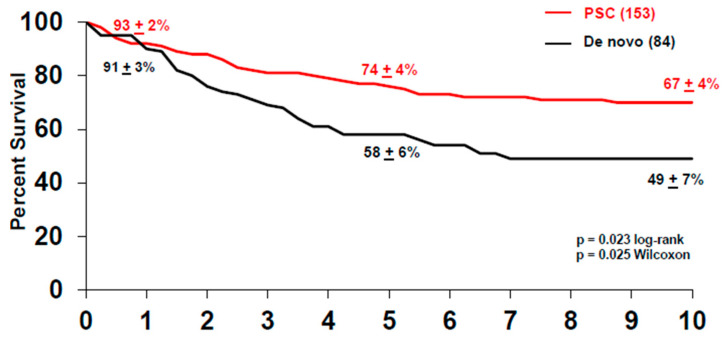
Survival after liver transplantation of patients with de novo (black) and primary sclerosing cholangitis (PSC)-associated pCCA (red). Patients who were transplanted for PSC-associated pCCA had better long-term survival when compared to patients transplanted for de novo pCCA.

**Figure 3 cancers-12-03157-f003:**
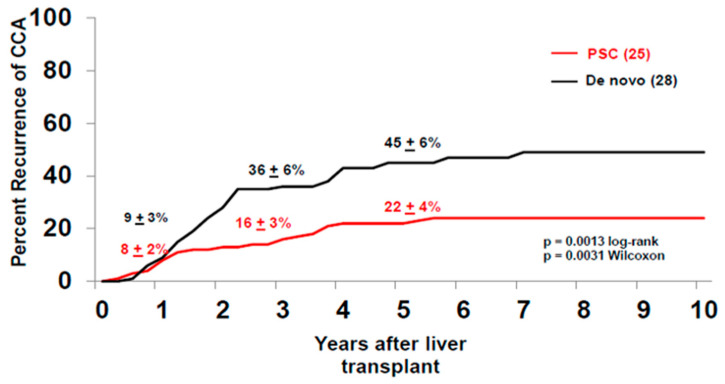
Tumor recurrence risk in percentage after liver transplantation in patients transplanted for de novo (black) and PSC-associated pCCA (red). A higher risk of tumor recurrence was observed in patients transplanted for de novo pCCA when compared to PSC-associated pCCA.

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
