# Peer review of "Selected Patients with Unresectable Perihilar Cholangiocarcinoma (pCCA) Derive Long-Term Benefit from Liver Transplantation"

_cancers, 2020, doi:10.3390/cancers12113157_

Round 1
Reviewer 1 Report
Good paper about Your experience!!!
Reviewer 2 Report
Fine with the improved manuscript and from my point of view now warrants publication in Cancers
This manuscript is a resubmission of an earlier submission. The following is a list of the peer review reports and author responses from that submission.
Round 1
Reviewer 1 Report
Pros and Cons: This manuscript summarized the potential strategy of liver transplantation for initially unresectable perihilar cholangiocarcinoma (pCCA). Neoadjuvant therapy following liver transplantation showed more prolonged survival and indicated a curative treatment for patients with pCCA. The authors have established and reported an excellent strategy for this challenging disease; however, there are several weaknesses in this manuscript.
1: For the discussion of the efficacy of neoadjuvant therapy, the patient selection bias should largely affect the analysis of survival benefit. The overall survival on intention-to-treat (ITT) analysis should also be added in this manuscript.
2: The therapeutic efficacy of neoadjuvant therapy is heterogeneous in patients with pCCA. CA19-9 alteration in the preoperative period and pathological evaluation for treatment efficacy in resected samples, such as Evans classification, are informative to judge the superior of this neoadjuvant protocol.
3: As you mentioned in this manuscript, the decrease of post-transplant recurrence is a key to more dramatically improve postoperative survival for de novo pCCA patients. For these patients, the sequenced systemic chemotherapy after chemoradiotherapy might boosts the inhibition of circulated micrometastases of cancer cells, since your protocol is likely to put much emphasis on local control by chemoradiotherapy and luck the power of systematic therapy, for example using gemcitabine and cisplatin. These two agents have been widely accepted as standard therapy and first choice for unresectable pCCA. You should add several discussions about this point.
Author Response
Comment #1: For the discussion of the efficacy of neoadjuvant therapy, the patient selection bias should largely affect the analysis of survival benefit. The overall survival on intention-to-treat (ITT) analysis should also be added in this manuscript.
Response: Thank you for bringing up this important point. We have added the ITT analysis as requested to the paper as Figure 1. The figure is as follows and is reference on page 9 of the revised manuscript:
|
|
|
|
Comment #2: The therapeutic efficacy of neoadjuvant therapy is heterogeneous in patients with pCCA. CA19-9 alteration in the preoperative period and pathological evaluation for treatment efficacy in resected samples, such as Evans classification, are informative to judge the superior of this neoadjuvant protocol.
Response: We have provided further clarification on the studies we cited in our review to discuss the predictive power of histopathologic characteristics the tumors on explants. The paper we cite in detail [1] in the revised manuscript developed robust criteria for the viable residual tumor, which they define as the estimate residual tumor (ERT) grade. This serves a similar purpose to the Evans classification which has been used to estimate residual tumor in post-treatment pancreatic resection specimens in pancreatic adenocarcinoma. The revised section can be found in pages 9 and 10 of the submitted manuscript. Unfortunately, the CA 19-9 is not a reliable marker of treatment response of the tumor given the frequency of subclinical and overt bacterial cholangitis in this stented population who develop further biliary tract strictures and cholangitis as a consequence of the radiation therapy.
Comment #3: As you mentioned in this manuscript, the decrease of post-transplant recurrence is a key to more dramatically improve postoperative survival for de novo pCCA patients. For these patients, the sequenced systemic chemotherapy after chemoradiotherapy might boosts the inhibition of circulated micrometastases of cancer cells, since your protocol is likely to put much emphasis on local control by chemoradiotherapy and lack the power of systematic therapy, for example using gemcitabine and cisplatin. These two agents have been widely accepted as standard therapy and first choice for unresectable pCCA. You should add several discussions about this point.
Response: We have not used this sequence of therapy, gemcitabine plus cisplatin after the chemoradiation therapy. We note there is little data to suggest efficacy of this regimen in pCCA. Indeed, in both the ABC-01 trial [2] and the PRODIGE-2-ACCORD18-UNICANCER GI trial [3], the confidence intervals crossed one indicating lack of efficacy for this subset of CCA. Hence, it would be an overstatement to state this is a standard of care. We do note we have used this regimen after transplant in patients with recurrent disease, where it has proved to be quite toxic and ineffective.

Reviewer 2 Report
This is a single center experience for liver transplantation in patients with unresectable perihilar cholangiocarcinoma which only can be done in highly experience centers and under a rigorous protocol with an expert multidisciplinary team.
I liked the study but I have some questions that might be completed:
- Over the last 25 years, have there been any changes about patterns of presentation?
- How many patients were cirrhotic in the De novo group?
- Is it possible to extend the information about the explants and how the histologic features affect the survival?
Author Response
Comment #1: Over the last 25 years, have there been any changes about patterns of presentation?
Response: There has not been a shift in the patterns of presentation as assessed by stage of disease which is quite restrictive to enter the protocol. We have not observed an increase in non-alcoholic fatty liver disease, nor other causes of chronic liver disease in this population.
Comment #2: How many patients were cirrhotic in the de novo group?
Response: This is an interesting question. There was one patient with alcoholic liver disease, 1 patient with HBV, 2 patients with HCV and 1 patient with HCV and HIV co-infection and 4 patients with NASH. Unfortunately, we do not have enough data to confirm the stage of their liver disease. This information is now included on page 8 of the revised manuscript.
Comment #3: Is it possible to extend the information about the explants and how the histologic features affect the survival?
Response: We have expanded our review of the pathologic assessments and their value in predicting outcomes on pages 10 and 11 which now reads as follows:
In particular, Lehrke et al closely examined extent of residual tumors and other histopathologic factors in predicting outcomes in a detailed study [24]. Residual tumor was estimated by a histologic assessment of the tumor bed area where the relative magnitude of the residual viable tumor to treatment-related necrosis and fibrosis was measured. The overall percentage of residual viable carcinoma was calculated as the average of estimated viable carcinoma percentages in all slides and were named as estimated residual tumor (ERT), and then classified into 4 categories: 1 (complete/near-complete response: ≤1% ERT), 2 (marked response: >1 to <10% ERT), 3 (moderate response: 10 to <30% ERT), and 4 (minimal response: ≥30% ERT). Category 1 included cases with no residual carcinoma or with only rare isolated tumor cells scattered in the tumor bed. Five year survival rate for patients for ERT 1 was 57%, vs 9% in ERT 4. It is important to note, however, that the absence of residual tumor on the explant does not eliminate the possibility of recurrence but does lower the risk [4], [5], [6], [1]. The histopathologic characteristics of the explant are also important in predicting tumor recurrence [4] [1]. In particular, the presence of perineural invasion was associated with a significant risk of recurrence after adjusting for confounding factors on multivariate analysis [1, 4]. Lehrke et al also examined the predictive power of the histologic grade of the residual tumor in the explant. The 4-tier grading system which is based on the percentage of glandular component of tumors was used to determine the histologic grades of the tumor (progressive glandular loss denotes loss of differentiation); grade 1, well differentiated; grade 2 moderately differentiated; grade 3, poorly differentiated; and grade 4, undifferentiated. Interestingly, the histologic grade of the residual tumor on the explant was a not a significant predictor of recurrence [1].
Reviewer 3 Report
Good review of Mayo Clinic experience
I would like to suggest:
- a brief but more incisive review on the radiological criteria on the diagnosis of cholangiocarcinoma (CCA) (li-rads), with a brief clarification of the role of ultrasound, ultrasound with contrast medium, CT scan and magnetic resonance
- a comment about Mayo Clinic experience (if present) about iCCA in cirrhotic patients
Author Response
Comment #1: a brief but more incisive review on the radiological criteria on the diagnosis of cholangiocarcinoma (CCA) (LI-RADS), with a brief clarification of the role of ultrasound, ultrasound with contrast medium, CT scan and magnetic resonance
Response: Thank you for the thoughtful question on the diagnosis of cholangiocarcinoma. Although LI-RADS criteria have been reported for intrahepatic CCA (iCCA); they have not been developed for pCCA. Indeed, pCCA usually presents as an obstructive biliary stricture (painless jaundice), with proximal bile duct dilatation without a mass lesion. As LI-RADS criteria are developed for well-defined mass lesions, it is unlikely that LI-RADS criteria for pCCA will ever be developed. As described in detail on pages 4, 5 and 6 of the revised manuscript, the diagnosis of pCCA relies heavily on a multi-disciplinary approach including cross-sectional imaging, ERCP, and conventional and advanced cytology testing,
Comment #2: a comment about Mayo Clinic experience (if present) about iCCA in cirrhotic patients
Response: Although we have a protocol in place (the Toronto Protocol) for patients with iCCA, we have not transplanted a patient on this protocol due to inability to obtain MELD exception points in the United States for this patient population.
REFERENCES
- Lehrke, H.D., et al., Prognostic Significance of the Histologic Response of Perihilar Cholangiocarcinoma to Preoperative Neoadjuvant Chemoradiation in Liver Explants. Am J Surg Pathol, 2016. 40(4): p. 510-8.
- Valle, J., et al., Cisplatin plus gemcitabine versus gemcitabine for biliary tract cancer. N Engl J Med, 2010. 362(14): p. 1273-81.
- Edeline, J., et al., Gemcitabine and Oxaliplatin Chemotherapy or Surveillance in Resected Biliary Tract Cancer (PRODIGE 12-ACCORD 18-UNICANCER GI): A Randomized Phase III Study. J Clin Oncol, 2019. 37(8): p. 658-667.
- Heimbach, J.K., et al., Predictors of disease recurrence following neoadjuvant chemoradiotherapy and liver transplantation for unresectable perihilar cholangiocarcinoma. Transplantation, 2006. 82(12): p. 1703-7.
- Darwish Murad, S., et al., Predictors of pretransplant dropout and posttransplant recurrence in patients with perihilar cholangiocarcinoma. Hepatology, 2012. 56(3): p. 972-81.
- Bhat, M., et al., Portal vein encasement predicts neoadjuvant therapy response in liver transplantation for perihilar cholangiocarcinoma protocol. Transpl Int, 2015. 28(12): p. 1383-91.